# CODEFUSION: A Pre-trained Diffusion Model for Code Generation

**Mukul Singh**
Microsoft
Delhi, India

**José Cambronero**
**Sumit Gulwani**
**Vu Le**
Microsoft
Redmond, US

**Carina Negreanu**
Microsoft Research
Cambridge, UK

**Gust Verbruggen**
Microsoft
Keerbergen, Belgium

## Abstract

Imagine a developer who can only change their last line of code—how often would they have to start writing a function from scratch before it is correct? Auto-regressive models for code generation from natural language have a similar limitation: they do not easily allow reconsidering earlier tokens generated. We introduce CODEFUSION, a pre-trained diffusion code generation model that addresses this limitation by iteratively denoising a complete program conditioned on the encoded natural language. We evaluate CODEFUSION on the task of natural language to code generation for Bash, Python, and Microsoft Excel conditional formatting (CF) rules. Experiments show that CODEFUSION (75M parameters) performs on par with state-of-the-art auto-regressive systems (350M–175B parameters) in top-1 accuracy and outperforms them in top-3 and top-5 accuracy, due to its better balance in diversity versus quality.

## 1 Introduction

Auto-regressive code generation models (Wang et al., 2021; Brown et al., 2020; Scholak et al., 2021; Feng et al., 2020; Fried et al., 2022) cannot easily reconsider tokens generated earlier in the decoding process. This limitation can lead to lower diversity generations (Lin et al., 2023) in the related domain of text. To balance diversity and quality of candidates generated, prior work has explored decoding strategies such as grouped beam search (Vijayakumar et al., 2018) or nucleus sampling (Holtzman et al., 2019).

Diffusion models, which have shown remarkable performance in image generation (Dhariwal and Nichol, 2021), have recently been extended to generate diverse text (Li et al., 2022; Lin et al., 2023). These approaches use an embedding layer to convert discrete tokens to continuous embeddings, where Gaussian noise can be added and predicted, to imitate the diffusion process. To map denoised embeddings back to discrete text, these approaches then select the vocabulary token with the closest embedding. In the code domain, where there are many syntactic and semantic constraints between tokens, independently projecting embeddings back to tokens can yield invalid programs.

We propose CODEFUSION, a natural language to code (NL-to-code) model that combines an encoder-decoder architecture (Raffel et al., 2020) with a diffusion process. The encoder maps the NL into a continuous representation, which is used by the diffusion model as an additional condition for denoising random Gaussian noise input. To generate syntactically correct code, we then feed the denoised embeddings to a transformer decoder, with full self-attention and cross attention with the embedded utterance, to obtain probability distributions over code tokens. Finally, we select the token with the highest probability at each index.

To pre-train CODEFUSION for code generation, we extend the continuous paragraph denoising (CPD) task introduced in Lin et al. (2023) to the code domain. Specifically, we only apply noise to tokens that correspond to identifiers in code or to built-in keywords in the target language. This denoising task allows the model to learn relations between critical code tokens (like variable names, function names and control flow built-ins).

We find that CODEFUSION yields more diverse code (higher $n$-gram fraction, lower embedding similarity, and higher edit distance) than auto-regressive models (see Table 2). The CPD objective, which biases the model towards learning to remove noise in a context-aware fashion, paired with a decoder that has access to the full denoised representation, jointly lead CODEFUSION to produce 48.5% more syntactically correct generations (averaged over three languages) when compared to GENIE, a text diffusion model (Table 3).

We evaluate CODEFUSION on NL-to-code for three different languages: Python (Yin et al., 2018), Bash (Lin et al., 2018), and conditional formatting

rules in Microsoft Excel (Singh et al., 2022). Our results show that CODEFUSION's (75M parameters) top-1 results are comparable or better than much larger state-of-the-art systems (350M–175B parameters). In top-3 and top-5, CODEFUSION performs better than all baselines.

This work makes the following contributions:

1. We propose CODEFUSION, the first diffusion-based NL-to-code model.

2. We adapt continuous paragraph denoising (CPD) to code and show that it substantially improves the results of CODEFUSION.

3. We compare CODEFUSION to auto-regressive code models and text diffusion models on the NL-to-code task in three languages.

## 2 Related Work

NL-to-code is a popular and challenging sequence to sequence problem. Previous techniques for NL-to-code have used RNNs (Brunner and Stockinger, 2021) and semantic parsers (Lee et al., 2021). Attention-based transformer models (Vaswani et al., 2017) have been adapted for the code domain, and include CodeBERT (Feng et al., 2020) (encoder only), T5 (Raffel et al., 2020) (encoder-decoder) and GPT-3 (Brown et al., 2020) (decoder only). More recently, large scale transformer-based models like CodeGen (Nijkamp et al., 2023) and TransCoder (Sun et al., 2023) have shown promising results. Instruction-tuning has also been used to further improve performance with models like StarCoder (Li et al., 2023a) and WizardCoder (Luo et al., 2023). These models can decode in an auto-regressive and non-auto-regressive fashion (Su et al., 2021). Recent surveys have discussed these models in more detail (Xu and Zhu, 2022; Niu et al., 2023; Xu et al., 2022).

Diffusion models have been popularly applied to unsupervised (Dhariwal and Nichol, 2021) and text conditioned image generation tasks (Saharia et al., 2022a,b; Ramesh et al., 2021). More recently, similar approaches have been extended to work in discrete domains such as text (Li et al., 2022; Lin et al., 2023; Li et al., 2023b; Reid et al., 2023).

## 3 Methodology

Figure 1 shows CODEFUSION's architecture. This section describes each component and our training and inference procedures.

### 3.1 Architecture

The input to CODEFUSION is a natural language utterance $s = \{s_1, s_2, \cdots, s_k\}$ and the output is a predicted code snippet $\hat{y} = \{\hat{y}_1, \hat{y}_2, \cdots, \hat{y}_s\}$. Both input and output are padded to a fixed dimension $n$. CODEFUSION has three main transformer-based components (an encoder $E$, a denoiser $N$, a decoder $D$) and a classification head $H$.

The transformer-based encoder transforms the tokenized utterance $s$ into a vector representation $E_s = E(s) = \{e_1, e_2, \cdots, e_n\}$.

Conditioned on the encoded utterance $E_s$ and the time $t$, the denoiser ($N$) predicts and removes noise $\epsilon_t$ from the noisy program embedding $x^t$ to obtain a predicted denoised program embedding $\hat{x}^0 = N(x^t, t, E_s)$. $N$ is a transformer block with cross-attention between $x^t$ and $E_s$ and full self-attention over $x^t$.

Before projecting the denoised embeddings back to discrete code tokens, we use a decoder ($D$), this time applying cross-attention to $\hat{x}^0$ and $E_s$, with full self-attention over $\hat{x}^0$, to compute a final hidden representation $D_s = \{d_1, d_2, \cdots, d_n\} = D(x^0, E_s)$. As opposed to prior text diffusion approaches, where tokens are generated independently, full self-attention allows each hidden dimension ($d_i$) to be generated with full information about the other dimensions.

Finally, $D_s$ is projected to actual code tokens with a classification head $H$ that computes a distribution over code tokens $p(y|d_i)$. We do not perform a search over these tokens and select $\hat{y}_i = \arg\max_y p(y|d_i)$ for each $i$.

### 3.2 Training

We train CODEFUSION in two phases: unsupervised pre-training of the denoiser and decoder on code snippets, and supervised fine-tuning of encoder, denoiser and decoder on (utterance, code snippet) pairs. Following prior work on diffusion for text, we use a trainable embedding layer $L$ to embed a code snippet $y$ into a continuous space where we can add (and remove) noise $\epsilon_t$ at timestep $t$.

We take inspiration from prior work on diffusion for text and adapt the loss from GENIE (Lin et al., 2023) to CODEFUSION by incorporating the hidden representation $D_s$ from the decoder. At time step $t$, the loss is computed as

$$\mathcal{L}_t = \|\hat{\epsilon}_t - \epsilon_t\| + \|D_s - L(y)\| - \log p(y|D_s)$$

and consists of three parts.

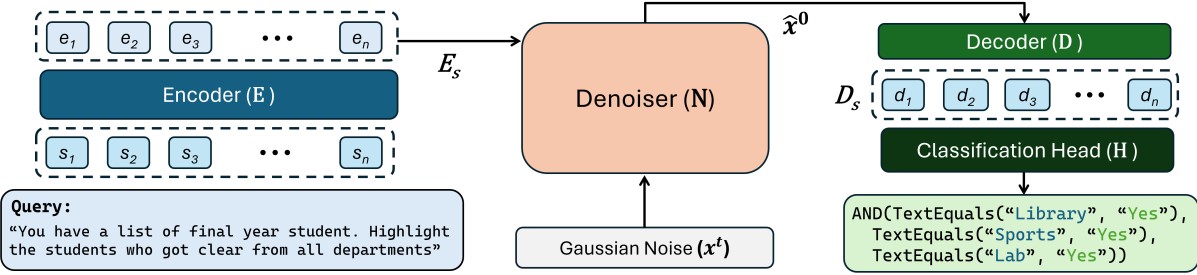

Figure 1: Architecture diagram for CODEFUSION showing the Encoder (E), Denoiser (N) and the Decoder (D) units.

1. We minimize the error between the predicted noise $\hat{\epsilon}_t$ and the actual noise $\epsilon_t$ to train $N$.

2. We minimize the error between the $D_s$ and embedded code to train $D$ and $L$.

3. We apply a standard cross-entropy loss over the outputs of the classification head, which produces predicted code tokens given $D_s$, and the ground truth code snippet $y$.

The loss function allows us to train the three main components of our model (denoiser, decoder and classification head) with a diffusion objective.

To pre-train the denoiser ($N$) and decoder ($D$) over a corpus of code snippets, we use two tasks: unsupervised code generation and our adaptation of continuous paragraph denoising (CPD) (Lin et al., 2023) for code. This code-specific CPD task only masks tokens associated with identifiers or built-in keywords from the target language. We randomly sample from these two tasks during pre-training.

Both pre-training and fine-tuning tasks use $\mathcal{L}_t$. Because there is no natural language utterance in pre-training, there is no input $E_s$ to the denoiser $N$. In the unsupervised code generation task, $E_s$ is replaced with Gaussian noise sampled at every denoising time step. In the CPD task, $E_s$ is computed by passing the masked code $y$ through encoder $E$.

### 3.3 Inference

During inference, we initialize $x^t$ with Gaussian noise and iteratively remove a (scheduler-determined) proportion of the noise over $T$ time steps to obtain $\hat{x}^0$ (Ho et al., 2020). During this iterative denoising, we do not use the decoder. After this iterative procedure, the decoder produces the final predicted code $\hat{y}$. We post-process $\hat{y}$ to select the tokens up to the first pad token.

## 4 Evaluation Setup

We briefly describe training, baselines, benchmarks and metrics. We provide further details for training and baselines in the Appendix.

### 4.1 Benchmarks

We evaluate CODEFUSION on NL-to-code for languages with varying complexity: Python, Bash, and conditional formatting (CF) rules in Microsoft Excel. The CoNaLa dataset (Yin et al., 2018) for Python consists of complex, multi-statement StackOverflow code snippets and associated NL questions. The Bash dataset (Lin et al., 2018) has complex, single-line Bash commands annotated with NL descriptions. The CF dataset (Singh et al., 2022) consists of Excel CF rules, which are single line programs of low complexity, annotated with NL. These benchmarks (and pre-training data—see next section) are made publicly available.[1]

### 4.2 Training

For our experiments, we instantiate the encoder ($E$) as a pre-trained CodeT5 encoder (Wang et al., 2021) (embedding dimension is 512), the denoiser ($N$) as a 10 layer transformer block, the decoder ($D$) as 6 transformer decoder layers, and the classification head ($H$) as a single fully connected layer.

In the training and pre-training phase, we use a square root noise schedule with 1200 diffusion steps (Wu et al., 2023). We use the tokenizer and vocabulary from CodeT5 (Wang et al., 2021) and target code length of 128 tokens. We use AdamW optimizer without weight decay (Loshchilov and Hutter, 2019) and a learning rate of 5e-4.

We pre-train the diffusion and decoder model on code snippets only. For Excel, we use a public corpus of 450K conditional formatting rules (Singh et al., 2022). For Python and Bash, we scrape

---

[1] https://github.com/microsoft/
prose-benchmarks/tree/main/CodeFusion

Table 1: Comparison of CODEFUSION with baselines on the task of NL to code generation for Python, Bash and CF rules. We report top-1, top-3 and top-5 predictions. **Model** denotes the underlying base model's checkpoint name. #P denotes the number of model parameters. We note the metric used for each language in parentheses.

| System description | | | Python (CodeBERT) | | | Bash (template) | | | CF Rule (execution) | | |
|---|---|---|---|---|---|---|---|---|---|---|---|
| **System** | **Model** | **#P** | top-1 | top-3 | top-5 | top-1 | top-3 | top-5 | top-1 | top-3 | top-5 |
| T5 | t5-large | 770M | 80.4 | 82.3 | 84.8 | 67.1 | 68.9 | 70.3 | 71.1 | 73.4 | 74.6 |
| CodeT5 | codet5-large | 770M | 80.5 | 83.1 | 85.0 | **67.6** | 69.3 | 70.5 | 72.7 | 75.3 | 75.8 |
| GPT-3 | text-davinci-003 | 175B | **82.5** | 83.7 | 85.8 | 66.9 | 67.7 | 68.4 | 70.3 | 72.4 | 72.8 |
| ChatGPT | gpt-3.5-turbo | Unknown | 80.6 | 82.5 | 83.9 | 66.1 | 66.9 | 67.8 | 70.8 | 73.1 | 74.5 |
| StarCoder | starcoder | 15.5B | 79.2 | 82.0 | 84.1 | 64.5 | 65.3 | 66.5 | 70.6 | 72.8 | 74.5 |
| CodeT5+ | codet5p-16b | 16B | 79.6 | 82.1 | 84.5 | 65.7 | 66.1 | 67.2 | 70.5 | 72.9 | 74.3 |
| CodeGen | codegen-350m | 350M | 80.1 | 81.8 | 83.7 | 67.2 | 69.2 | 70.3 | 71.4 | 73.7 | 75.0 |
| Diffusion-LM | Custom | 50M | 70.4 | 74.3 | 76.5 | 59.4 | 61.6 | 62.0 | 62.4 | 65.5 | 68.2 |
| GENIE | Custom | 93M | 73.2 | 77.1 | 80.3 | 60.0 | 61.5 | 62.3 | 62.9 | 66.8 | 68.7 |
| **CODEFUSION** | **Custom** | 75M | 80.7 | **86.3** | **90.3** | 66.7 | **70.2** | **72.0** | **72.8** | **76.7** | **78.5** |

GitHub notebooks and StackOverflow posts with tags python, bash and powershell, using a regex extractor to detect code (Lin et al., 2018).

### 4.3 Baselines

We use a combination of transformer and text diffusion models as baselines: T5 (Raffel et al., 2020), GPT-3 (text-davinci-003) (Brown et al., 2020), ChatGPT (gpt-3.5-turbo) (OpenAI, 2023), CodeT5 (Wang et al., 2021), StarCoder (Li et al., 2023a), CodeT5+ (Wang et al., 2023), CodeGen (Nijkamp et al., 2023), Diffusion-LM (Li et al., 2022), and GENIE (Lin et al., 2023). Out of these GPT-3, ChatGPT, StarCoder, CodeT5+ are used in an in-context learning setting with five examples dynamically selected using SentenceBERT (Reimers and Gurevych, 2019) similarity of NL utterance, the others are fine-tuned on our datasets. Diffusion-LM and GENIE are also pre-trained on our corpus.

### 4.4 Metrics

We evaluate Bash generation using the template match metric—which performs some basic normalization—provided with the dataset. We evaluate Python using CodeBERTScore (Zhou et al., 2023), which has been shown to be a high quality non-execution-based code matching metric. We evaluate CF using execution match (Singh et al., 2022) by executing a rule on the data column and comparing to the expected output.

To evaluate generation diversity, we measure (1) count of distinct token n-grams in the generated code divided by number of tokens (Vijayakumar et al., 2018), (2) summary statistics over pairwise similarities of CodeBERT encodings (Perlitz et al., 2023), and (3) summary statistics over pairwise

string edit distances (Perlitz et al., 2023).

## 5 Evaluation

We investigate the following questions. **Q1.** Does CODEFUSION generate correct and diverse code? **Q2.** How do different design decision impact performance? **Q3.** How does the latent representation evolve during the diffusion steps?

### 5.1 Performance and Diversity (Q1)

Table 1 summarizes performance in top-1, top-3 and top-5 settings for CODEFUSION and baselines.

In top-1, CODEFUSION performs on par with or even better than (much larger) auto-regressive models. For Python, only GPT-3 (175B) performs better than CODEFUSION (75M). In top-3 and top-5, CODEFUSION outperforms all baselines, consistent with previous observations that auto-regressive models with high top-1 performance sacrifice diversity in their generations (Poesia et al., 2022).

Table 2 shows diversity results averaged across all benchmark tasks, over the top-5 generations for each model, for CODEFUSION and auto-regressive (T5, CodeT5, StarCoder, CodeGen, GPT-3) baselines. CODEFUSION produces generations of higher diversity compared to auto-regressive models.

Like CODEFUSION, other diffusion methods (Diffusion-LM and GENIE) improve for top-3 and top-5 relative to top-1. They fall short of CODEFUSION as a result of generating syntactically invalid programs. Table 3 shows the fraction of syntactically valid generations for CODEFUSION and diffusion baselines. CODEFUSION generations are more often syntactically valid compared to diffusion models not designed for code: 33.8% more

Table 2: Comparison of diversity in top-5 code generations for CODEFUSION and baselines for Python, Bash and CF rules. We report fraction of distinct token-level **n-grams**, pairwise similarities of CodeBERT **embedding** of outputs, and statistics over pairwise string normalized **edit distance** of outputs.

| Models | N-grams (↑ better) | | | | Embedding similarity (↓ better) | | | Edit distance (↑ better) | | |
|---|---|---|---|---|---|---|---|---|---|---|
| | **1** | **2** | **3** | **4** | **Min** | **Max** | **Mean** | **Min** | **Max** | **Mean** |
| T5 | 0.29 | 0.45 | 0.54 | 0.6 | 0.93 | 0.99 | 0.97 | 0.08 | 0.58 | 0.36 |
| CodeT5 | 0.27 | 0.41 | 0.46 | 0.51 | 0.94 | 0.99 | 0.97 | 0.08 | 0.56 | 0.32 |
| GPT-3 | 0.24 | 0.36 | 0.42 | 0.48 | 0.97 | 0.99 | 0.99 | 0.00 | 0.28 | 0.21 |
| StarCoder | 0.32 | 0.48 | 0.56 | 0.61 | 0.89 | 0.98 | 0.96 | 0.11 | 0.61 | 0.39 |
| CodeGen | 0.28 | 0.41 | 0.49 | 0.55 | 0.94 | 0.99 | 0.97 | 0.09 | 0.52 | 0.34 |
| **CODEFUSION** | **0.53** | **0.65** | **0.74** | **0.81** | **0.83** | **0.97** | **0.91** | **0.21** | **0.83** | **0.57** |

Table 3: % of top-1 generations that are syntactically-valid for CODEFUSION and text diffusion-based baselines. CODEFUSION generates more valid candidates.

| System | Python | Bash | CF Rules |
|---|---|---|---|
| Diffusion-LM | 19.5 | 40.4 | 74.3 |
| GENIE | 24.2 | 54.2 | 78.6 |
| CODEFUSION | **67.6** | **73.4** | **94.5** |

versus Diffusion-LM and 26.2% more versus GE-NIE averaged across all three languages.

## 5.2 Ablations (Q2)

Table-4 shows the results of CODEFUSION with various changes. Removing either pre-training task significantly reduces performance (-10.9% for code generation and -4.6% for CPD on average across the three languages). Results by replacing $D$ and $H$ with grounding (pick closest vocabulary token at last denoising step) or clamping (pick closest vocabulary token at each denoising step) highlights the benefit of using a decoder before rounding.

Table 4: Ablations for CODEFUSION (M). "–" means the component is removed. Full model beats all ablations.

| Model | Python | Bash | CF Rule |
|---|---|---|---|
| M – Code Generation | 70.9 | 52.3 | 64.2 |
| M – CPD Objective | 76.7 | 61.1 | 68.2 |
| M with Grounding | 73.5 | 60.5 | 63.2 |
| M with Clamping | 77.1 | 63.2 | 64.4 |
| **M (CODEFUSION)** | **80.7** | **66.7** | **72.8** |

## 5.3 Gradual Refinement (Q3)

We study how CODEFUSION gradually reached the final result. For this experiment, we stop the denoising at a timestep $t \in [0, T]$, and generate a code snippet for the current state. We measure the normalized string edit distance obtained at each time step (in increments of 100 steps). Figure 2 shows

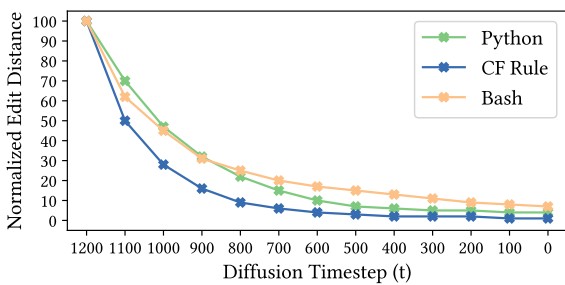

Figure 2: Average normalized edit distance for CODEFU-SION generations against increasing diffusion timesteps.

```
NL:     Copy the content of file 'file.txt' to file 'file2.txt'
Target: shutil.copy('file.txt', 'file2.txt')

t = 1200  Ss ec between Useful 144 Location copyright pen they destinat
          Path caret corev adapter NameAnd Ǵiterating Ǵface Ǵpass ǴFH

t = 1000  (.* for copyInt filefiletxt, filefiletxt queryResult SE Screen

t = 800   destutil..copyInt'filetxt', 'filetxt') return 'filetxt'

t = 600   destutil.copyInt'file.txt', 'fi2.txt')

t = 400   shutil.copyInt'file.txt', 'file.txt')

t = 200   shutil.copy('file.txt', 'file2.txt')

t = 0     shutil.copy('file.txt', 'file2.txt')
```

Figure 3: Successive stages of denoising by CODEFU-SION on an example from the Python benchmarks.

that the edit distance decreases with $t$. The drop is much faster for CF rules as these are simpler to generate than full programs and bash commands. An example visualizing this is shown in Figure 3.

## 6 Conclusion

We propose CODEFUSION, the first diffusion natural language to code (NL-to-code) generation model. With a decoder and code-focused pre-training, CODEFUSION generates more syntactically correct programs than existing text diffusion models and more diverse programs than existing auto-regressive code models. Our evaluation shows that CODEFUSION competes with state-of-the-art transformer code generation models on Python, Bash and Excel conditional formatting rules.

# 7   Limitations

CODEFUSION is not a global system as we only consider natural language utterances in English. Furthermore, natural language specifications can be provided at varying levels of detail – utterances with less detail may result in worse performance. We consider various programming languages, but more complex languages may result in worse performance. We also find that CODEFUSION struggles when tasked with generating longer code snippets or programs that have long-ranged syntactic dependencies. Because diffusion-based models denoise iteratively, inference latency is substantial, rising quadratically with target generation length.

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

## A  Implementation

We first describe the environment used to conduct experiments and then describe parameters for each component and training in detail.

### A.1  Hardware Specifications

All experiments and studies have been carried out on Python software (version 3.8.7). The system used to run the experiments uses an Intel Core i7 processor (base at 1.8 GHz) along with 4 V100 GPU units, a 64-bit operating system, and 56 GB RAM. CODEFUSION took 8 hours to pre-train and 3 hours to fine-tune on average for each dataset.

### A.2  Training Parameters

In the training and pre-training phase, we use a square root noise schedule with 1200 diffusion steps (Wu et al., 2023). We use the tokenizer and vocabulary from CodeT5 (Wang et al., 2021). We use a learning rate of 5e-4m with a batch size of 64 and a target length of 128. Further, we use AdamW optimizer without weight decay (Loshchilov and Hutter, 2019).

### A.3  Pre-training Data

Table 5 shows the breakdown of the number of samples used for pre-training along with their average length in terms of number of tokens.

Table 5: Summary of dataset used to pre-train CODE-FUSION. We report the number of samples and average length of each code snippet for all languages.

| Language | Samples | Avg Length |
|---|---|---|
| Python | 56K | 78.4 |
| Bash | 35K | 45.3 |
| Conditional Formatting Rules | 100K | 23.4 |

## B  Baselines

We describe imeplementation details of baseline systems.

### B.1  T5, CodeT5 and CodeGen

T5 is the base model of CODEFUSION and uses an encoder-decoder architecture. CodeT5 uses T5 as the base model and pre-trains it on code generation and understanding tasks. We train both T5 and CodeT5 to generate code from the input utterance. The input to T5 is same as that of CODEFU-SION fine-tuning (section-3.2). The models are optimized on standard cross-entropy loss with Adam optimizer at $1e - 4$ learning rate for 100 epochs. We find that CodeT5 performs better than T5. This is due to CodeT5 being pre-trained for code generation. These are trained on the train split of the datasets and evaluated on the test split. For decoding, we use beam search with beam size of 5.

### B.2  GPT-3, ChatGPT, StarCoder and CodeT5+

We use the pre-trained version of GPT-3 (Text DaVinci 003), ChatGPT (gpt-3.5-turbo), StarCoder (bigcode/starcoder) and CodeT5+ (codet5p-16b) models without fine-tuning in a few shots setting. We use 5 examples in the prompt. The few-shot examples are selected from the training corpus based on cosine similarity of sentence-BERT (Reimers and Gurevych, 2019) embedding of the query. For GPT-3 and ChatGPT, we experiment with temperature values between 0 to 1 with a step of 0.1 and report the best result with all other parameters at default value. For StarCoder and CodeT5+ we use beam search with beam size 5.

### B.3  GENIE

We implement GENIE as described in (Lin et al., 2023). We set the diffusion timestep $T = 1200$, embedding dimension to 256 and encoding and generation length to 128. We choose these parameters to be consistent with CODEFUSION. We also pretrain on the same corpus used to pretrain CODE-FUSION. For sampling top-k candidates, we use the same inference algorithm highlighted by authors in (Lin et al., 2023).

### B.4  Diffusion-LM

We implement Diffusion-LM as described in (Li et al., 2022). We set the diffusion timestep $T = 1200$, embedding dimension to 256 and encoding and generation length to 128, similar to GENIE. We use the controlled generation classifier to convert the text-to-code generation to a completion task. We simply extend the infilling task, the authors propose for text generation (Li et al., 2022).

## C  Further Results

In this section we describe additional results and examples from our corpus.

### C.1  Exact Match Results

We also show the exact match accuracy for CODE-FUSION and baselines on the benchmarks. TABLE 6

Table 6: Comparison of CODEFUSION with baselines on the task of text to code generation for Python, Bash and CF rules. We report top-1, top-3 and top-5 exact code match of the predictions. "Model" column denotes the underlying base model's checkpoint name. #P denotes the number of model parameters.

| System description | | | Python | | | Bash | | | CF Rule | | |
|---|---|---|---|---|---|---|---|---|---|---|---|
| **System** | **Model** | **#P** | **top-1** | **top-3** | **top-5** | **top-1** | **top-3** | **top-5** | **top-1** | **top-3** | **top-5** |
| T5 | t5-large | 770M | 5.2 | 6.1 | 6.7 | 13.5 | 14.7 | 15.2 | 63.1 | 65.2 | 67.6 |
| CodeT5 | codet5-large | 770M | 5.5 | 6.4 | 7.1 | 14.1 | 14.9 | 15.5 | 63.2 | 65.7 | 67.8 |
| GPT-3 | text-davinci-003 | 175B | **7.5** | **8.2** | 8.8 | 12.9 | 13.7 | 14.4 | 60.8 | 61.4 | 62.7 |
| ChatGPT | gpt-3.5-turbo | Unknown | 5.6 | 6.2 | 6.5 | 12.0 | 12.6 | 12.9 | 62.9 | 65.0 | 67.6 |
| StarCoder | starcoder | 15.5B | 4.8 | 6.0 | 6.4 | 12.1 | 12.5 | 13.4 | 62.6 | 64.8 | 66.9 |
| CodeT5+ | codet5p-16b | 16B | 4.9 | 6.1 | 6.6 | 12.3 | 12.8 | 13.9 | 62.8 | 64.9 | 67.0 |
| CodeGen | codegen-350m | 350M | 5.0 | 5.9 | 6.3 | **13.6** | 14.9 | 15.2 | 63.3 | 65.4 | 67.7 |
| DiffusionLM | Custom | 50M | 1.4 | 2.3 | 2.8 | 7.4 | 8.6 | 9.0 | 48.8 | 50.4 | 53.1 |
| GENIE | Custom | 93M | 1.7 | 2.5 | 3.0 | 8.0 | 9.5 | 10.3 | 49.5 | 52.4 | 54.6 |
| **CODEFUSION** | **Custom** | 75M | 5.1 | 7.2 | **9.0** | 13.5 | **15.3** | **16.4** | **63.4** | **67.6** | **69.1** |

```
        NL: | Copy the content of file 'file.txt' to file 'file2.txt'
    Target: | shutil.copy('file.txt', 'file2.txt')
-----------------------------------------------------------------------
  t = 1200  | Ss ec between Useful 144 Location copyright pen they destinat
            | Path caret corev adapter NameAnd Ġiterating Ġface Ġpass ĠFH

  t = 1000  | (.* for copyInt filefiletxt, filefiletxt queryResult SE Screen

   t = 800  | destutil..copyInt'filetxt', 'filetxt') return 'filetxt'

   t = 600  | destutil.copyInt'file.txt', 'fi2.txt')

   t = 400  | shutil.copyInt'file.txt', 'file.txt')

   t = 200  | shutil.copy('file.txt', 'file2.txt')

     t = 0  | shutil.copy('file.txt', 'file2.txt')
```

Figure 4: Figure showing the various stages of denoising in CODEFUSION on an example from the Python benchmarks where CODEFUSION succeeds. CODEFUSION starts from pure noise and gradually denoises to generate the target code.

shows these results. CODEFUSION performs comparable to transformer based models and better than other diffusion based text generation approaches for top-1 accuracy. For top-3 accuracy, CODEFUSION outperforms all baselines. This is consistent with the results in Table 1 and show that CODEFUSION produces better and more diverse candidate programs for a variety of tasks.

## C.2 Visualizing Diffusion

CODEFUSION iteratively denoises the latent representation to construct the final target. This can be visualized by mapping the representation at each time step to discrete tokens. We follow the setup as explained in Section-5.3. Figure 4 shows a success example from the Python benchmarks where CODEFUSION is able to generate the correct code. The figures shows the input query, the target code

and the reconstructed output from CODEFUSION at timesteps $t = \{200, 400, 600, 800, 1000, 1200\}$. We can see how CODEFUSION gradually denoises and generates the correct code. Figure 5 shows a failure case where CODEFUSION is unable to generate the correct code. The user asks to remove the directory tree *'folder_name/'*. CODEFUSION's generation is incorrect as os.removedir is not a valid function, the correct function name is os.removedirs. Further, this function only removes empty directories while the user wanted to remove the directory tree which includes files.

## C.3 Effect of Diffusion Time Step

The number of diffusion timesteps is directly related to the generation quality as shown in (Saharia et al., 2022a). We explore how CODEFUSION is affected by the number of timesteps

| | |
|---|---|
| **NL:** | remove directory tree '/folder_name' |
| **Target:** | shutil.rmtree('/folder_name') |

| | |
|---|---|
| **t = 1200** | `Ġcidr 00000 CallWithMethodType ValueList $subdomain Ġpow Ġ{' ĠtagName Ġdoct` |
| **t = 1000** | `des.os.(*rem# ('/foldername', file.rem(( os.del f.read() open as subdomain` |
| **t = 800** | `des.os.(*rem# ('/foldername',  file.rem(( os.del f.read() open -` |
| **t = 600** | `oshut.removetr#('/folder', '/name').rem((` |
| **t = 400** | `os.removetr#('folder name', file)` |
| **t = 200** | `os.removeditr('/folder name')` |
| **t = 0** | `os.removedir('/folder_name')` |

Figure 5: Figure showing the various stages of denoising in CODEFUSION on an example from the Python benchmarks where CODEFUSION fails. CODEFUSION starts from pure noise and gradually denoises to generate the target code. The final generation is incorrect here as the correct function name is `os.removedirs` and also this function only removes empty directories while the user wanted to remove directory with files.

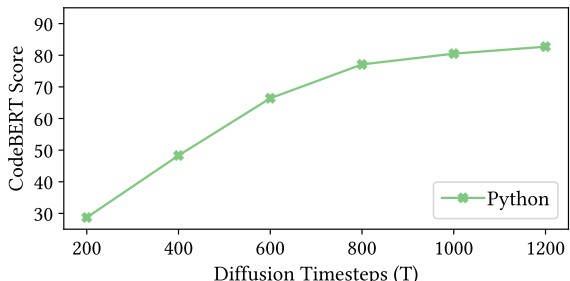

Figure 6: Plot showing the top-1 CodeBERT score for CODEFUSION for Python against increasing diffusion timesteps. Performance improves with increasing timesteps and stabilizes at $T = 1000$

for Python. We try different timestep values, $T = \{200, 400, 600, 800, 1000, 1200\}$ and plot the CodeBERT score corresponding to each variation. Figure 6 show the CodeBERT score against increasing timesteps. We see that the dependence of quality with timesteps is true for CODEFUSION as well. Adding timesteps has a diminishing gain as we see the plot flatten at $t = 1000$. It should be noted that adding timesteps also increases the inference latency and memory requirements of the model.

### C.4 Latency and Memory

Diffusion models are known to be complex with millions of parameters, and have higher latency and memory requirements than transformer based models. This is due to repeated sampling and sequential denoising operations. CODEFUSION has 75 Million parameters and requires a disk space of 544 Mega Bytes. The average inference latency on

the benchmarks was found to be 2318 milliseconds. The average GPU memory used was 928 Mega Bytes and the maximum GPU memory used was 1042 Mega Bytes.

## D   Background

### D.1   Transformer based Sequence Generation

Transformer based language models (Vaswani et al., 2017) are conditional generative models implemented through auto-regressive (AR) decoding. These models predict the likelihood of the target token $y_t$ using the conditional input encoding and previously generated tokens $y_1, y_2, \cdots, y_{t-1}$. The likelihood of the generated sequence is given by:

$$\mathcal{P}(y|x) = \prod_{i=1}^{N} p(y_i|y_{1:i-1}; x) \qquad (1)$$

### D.2   Diffusion Model

Diffusion processes are a discrete-time Markov process. The process starts with an initial state $x_0$ at timestep $t = 0$, where $x_0$ is from the original data distribution. The process moves forward by gradually adding Gaussian noise to $x_0$ in accordance to the variance schedule $\beta_1, \cdots, \beta_T$. Since the forward process only adds noise based on a schedule, at any timestep $t + 1$, $x_{t+1}$ can be expressed in terms of $x_t$ as

$$q(x_{t+1}|x_t) = \mathcal{N}\left(x_{t+1}; \sqrt{1 - \beta_{t+1}}x_t, \beta_{t+1}I\right) \quad (2)$$

During training, a diffusion model learns to perform the inverse diffusion process, wherein it predicts the noise at the current state, $x_t$ at timestep $t$.

By using the predicted noise, we can generate the previous state, $x_{t-1}$ by subtracting the noise from $x_t$ and rescaling the mean. Thus, the distribution of $x_{t-1}$ given $x_t$ is simply a Gaussian with mean $\mu_\theta^{t-1}$ and variance $\sigma_\theta^{t-1^2}$ where, $\theta$ are the parameters of the neural network.

The diffusion model is trained by minimizing the mean squared error between the true mean $\mu_\theta^{t-1}$ and the predicted mean $\hat{\mu}_\theta^{t-1}$.

### D.3 Diffusion Models for Text Generation

Recent works in text generation have started exploring diffusion models. (Li et al., 2022) proposes a system which uses embeddings to convert discrete tokens in text to continuous representations which form the target distribution. The diffusion model is then trained to denoise random Gaussian noise gradually to generate a sample from the target distribution. (Lin et al., 2023) extends this to conditional text generation by encoding the input and concatenating it with the diffusion state. (Wu et al., 2023) introduces clamping to force the model to follow the target distribution at each timestep, to improve performance.