# OpenReview forum: "CodeFusion: A Pre-trained Diffusion Model for Code Generation"
_EMNLP/2023/Conference — EMNLP 2023 Main_

### Official Review · Reviewer_XjFk · 2023-07-25

**Typos Grammar Style And Presentation Improvements:** 1. Authors might consider using the a…
**Soundness:** 4

**Excitement:**

4: Strong: This paper deepens the understanding of some phenomenon or lowers the barriers to an existing research direction.

**Missing References:**

Please refer to "Typos Grammar Style And Presentation Improvements."

**Paper Topic And Main Contributions:**

This paper proposes CODEFUSION that applies diffusion LMs to the domain of neural code intelligence, aiming for diverse and reliable code generation. Specifically, the authors combine transformer architecture and diffusion techniques and then leverage CPD tasks to curb the model from generating syntactically invalid programs. CODEFUSION is evaluated on the tasks of NL2Code generation tasks for Bash, Python, and Excel Conditional Formatting (CF) rules, and can reach comparable performance to mainstream auto-regressive LMs. Moreover, benefiting from the randomness of diffusion LMs, the proposed method reaches higher top-3/5 performance compared to other strong baselines.

**Reasons To Accept:**

1. The paper successfully leverages diffusion LMs to conduct code-related tasks and becomes the first diffusion-based NL2Code model.
2. Authors empirically demonstrate the effectiveness of CODEFUSION by solid evaluations on NL2Bash, Python code generation, and Excel Conditional Formatting tasks.
3. As a short paper, the paper includes sufficient analysis and ablations of the results and provides an in-depth discussion of diversity and refinement.
4. The overall presentation of this paper reaches the standards of an *ACL publication.

**Reasons To Reject:**

The comparisons with other LMs such as text-davinci-003 and CodePTMs like CodeT5 might be slightly “unfair”. The reason is that these baselines are general-purpose models, which can support generating different PLs by a single checkpoint (and can to some extent perform zero-shot). However, as is provided in A.3, CODEFUSION is task-specifically trained.

**Reproducibility:**

5: Could easily reproduce the results.

**Reviewer Confidence:**

5: Positive that my evaluation is correct. I read the paper very carefully and I am very familiar with related work.

---

> ### Author Rebuttal · Authors · 2023-08-29
>
> **Question-1: The comparisons with other LMs such as text-davinci-003 and CodePTMs like CodeT5 might be slightly “unfair”. The reason is that baselines are general-purpose models, which can support generating different PLs by a single checkpoint (and can to some extent perform zero-shot)**
>
> We fine-tuned CodeT5 and used text-davinci-003 in a few-shot prompting (as is common in literature). We have now also extended our evaluation to include results with CodeGen [1], StarCoder [2], CodeT5+ [3] and ChatGPT/GPT-3.5 [4]. The extended table is shown below for the Python (CoNaLa) Dataset:
>
> | System       | Model   | Param | Top-1 | Top-3 | Top-5 |
> |--------------|---------|-------|-------|-------|-------|
> | T5           | T5      | 770M  | 80.4  | 82.3  | 84.8  |
> | CodeT5       | T5      | 770M  | 80.5  | 83.1  | 85    |
> | GPT-3        | GPT 3   | 175B  | 82.5  | 83.7  | 85.8  |
> | ChatGPT      | GPT 3.5 | 20B   | 80.6  | 82.5  | 83.9  |
> | StarCoder    | GPT-2   | 15.5B | 79.2  | 82    | 84.1  |
> | CodeT5+      | T5      | 3B    | 78.6  | 81.1  | 83.5  |
> | CodeGen      | Custom  | 350M  | 80.1  | 81.8  | 83.7  |
> | Diffusion-LM | Custom  | 50M   | 70.4  | 74.3  | 76.5  |
> | GENIE        | Custom  | 93M   | 73.2  | 77.1  | 80.3  |
> | CODEFUSION   | T5      | 75M   | 80.7  | 86.3  | 90.3  |
>
>  \
> **Question-2: I suggest adding more 2022/23 papers about models and code generation into references to keep the paper up-to-date**
>
> We will extend our related work discussion to reflect on new work. Specifically, we will add [1,2,3,4,5,6,7,8,9,10,11,12,13,14,15,16] to the related works and update evaluation to include [1,2,3,4].\
> The new evaluation table is attached in Question-1.
>  \
>  \
>  \
> **Question-3: Authors might consider using the additional one page to add more details of the model training and diffusion LMs (like the paragraphs in the Appendix) if accepted.**
>
> Please see the link below for our source code. We include this for review purposes only. We will extend the appendix with more details on training and dataset. In addition, we will be publicly releasing our benchmark tasks and our model’s generations.
>
> [Source Code](https://figshare.com/s/1ac0b38ebf9ddc6a4a46)
>  \
>  \
>  \
> **References**
>
> [1]: Nijkamp, Erik, et al. "Codegen: An open large language model for code with multi-turn program synthesis." arXiv preprint arXiv:2203.13474 (2022).
>
> [2]: Li, Raymond, et al. "StarCoder: may the source be with you!" arXiv preprint arXiv:2305.06161 (2023).
>
> [3]: Wang, Yue, et al. "Codet5+: Open code large language models for code understanding and generation." arXiv preprint arXiv:2305.07922 (2023).
>
> [4]: https://openai.com/blog/chatgpt
>
> [5]: Vijayakumar, Ashwin K., et al. "Diverse beam search: Decoding diverse solutions from neural sequence models." arXiv preprint arXiv:1610.02424 (2016).
>
> [6]: Perlitz, Yotam, et al. "Diversity enhanced table-to-text generation via type control." arXiv preprint arXiv:2205.10938 (2022).
>
> [7]: Lomshakov, V., Kovalchuk, S., Omelchenko, M., Nikolenko, S., Aliev, A. (2023). Fine-Tuning Large Language Models for Answering Programming Questions with Code Snippets. In: Mikyška, J., de Mulatier, C., Paszynski, M., Krzhizhanovskaya, V.V., Dongarra, J.J., Sloot, P.M. (eds) Computational Science – ICCS 2023. ICCS 2023. Lecture Notes in Computer Science, vol 14074. Springer, Cham. https://doi.org/10.1007/978-3-031-36021-3_15
>
> [8]: Soliman, Ahmed S., Mayada M. Hadhoud, and Samir I. Shaheen. "MarianCG: A code generation transformer model inspired by machine translation." Journal of Engineering and Applied Science 69.1 (2022): 1-23.
>
> [9]: Beau, Nathanaël, and Benoît Crabbé. "The impact of lexical and grammatical processing on generating code from natural language." arXiv preprint arXiv:2202.13972 (2022).
>
> [10]:	Norouzi, Sajad, Keyi Tang, and Yanshuai Cao. "Code Generation from Natural Language with Less Prior and More Monolingual Data." arXiv preprint arXiv:2101.00259 (2021).
>
> [11]:	A Systematic Evaluation of Large Language Models of Code https://arxiv.org/abs/2202.13169
>
> [12]:	A Survey on Pretrained Language Models for Neural Code Intelligence https://arxiv.org/abs/2212.10079
>
> [13]:	An Empirical Comparison of Pre-Trained Models of Source Code https://arxiv.org/abs/2302.04026
>
> [14]:	Coder Reviewer Reranking for Code Generation https://arxiv.org/abs/2211.16490
>
> [15]:	TransCoder: Towards Unified Transferable Code Representation Learning Inspired by Human Skills https://arxiv.org/abs/2306.07285
>
> [16]:	WizardCoder: Empowering Code Large Language Models with Evol-Instruct https://arxiv.org/abs/2306.08568

---

### Official Review · Reviewer_ADn5 · 2023-08-03

**Soundness:** 3

**Excitement:**

4: Strong: This paper deepens the understanding of some phenomenon or lowers the barriers to an existing research direction.

**Missing References:**

Missing lots of CodeLLMs appeared in 2022/23, like CodeGen [1] and CodeX [2], etc.

[1] "Codegeex: A pre-trained model for code generation with multilingual evaluations on humaneval-x." arXiv preprint arXiv:2303.17568 (2023).

[2] "Codegen: An open large language model for code with multi-turn program synthesis." arXiv preprint arXiv:2203.13474 (2022).

**Paper Topic And Main Contributions:**

To enable code generation diversity and maintain the validity of generated code, the author introduce CODEFUSION, a novel pre-trained diffusion code generation model that employs an attention-based decoding strategy and a mask-denoising objective to generate code, while attending to contextual token embeddings and sequential dependencies.  The result shows CODEFUSION can compete with or outperform existing transformer and diffusion-based text generation models on the task of text to code generation for Python, Bash and formatting rules.

**Questions For The Authors:**

1. In Figure 1, how is the Gaussian noise added to the encoder? Why is there no connection between the Gaussian noise and the representations?

2. I did not quite understand line 153. How does the formula in line 149 allow each token to be generated with full information about the other tokens?

3. Could you provide a more detailed explanation as to why this design can generate code with correct syntax and diversity?

**Reasons To Accept:**

1. CODEFUSION is the first diffusion-based NL-to-code model, which brings a novel approach.

2. The authors have clearly defined their formulas.

3. The experimental results are impressive.

**Reasons To Reject:**

1. The abstract could benefit from additional clarity to better convey the research motivation and objectives. The current statement, "Diffusion models have been extended to discrete domains like text, but have not yet been successfully applied to code tasks," falls short in providing a clear rationale for the study. It would be helpful to specify the particular question and its importance in code generation, such as "increase the diversity of code generation".
In the Introduction, you mention that "Transformer-based NL-to-Code systems can struggle to produce diverse generations." While this is a valid point, it should be noted that existing large code models, like in CodeGeex [1], mentioned "We use budget allocation methods to help improve the diversity of the generated solutions.”, which have shown significant progress in this aspect. If the aim of your work is to improve generation diversity, I suggest elaborating on how your research differs from these existing studies and what unique contributions it is expected to make.

2. How does the design of the loss function, as described in line 171, ensure the diversity of the generated code? Why don't the second and third terms of the Loss function require weight parameters like \alpha or \beta? Moreover, the lack of supporting code in this paper does not facilitate the reproducibility of the feasibility of this work.

3. I recommend that you explore more large-scale code models such as CodeGen [2] and CodeX [3]. However, this is a short paper, so this is just a suggestion.

[1] "Codegeex: A pre-trained model for code generation with multilingual evaluations on humaneval-x." arXiv preprint arXiv:2303.17568 (2023).

[2] "Codegen: An open large language model for code with multi-turn program synthesis." arXiv preprint arXiv:2203.13474 (2022).

[3] "Evaluating large language models trained on code." arXiv preprint arXiv:2107.03374 (2021).

**Reproducibility:**

4: Could mostly reproduce the results, but there may be some variation because of sample variance or minor variations in their interpretation of the protocol or method.

**Reviewer Confidence:**

4: Quite sure. I tried to check the important points carefully. It's unlikely, though conceivable, that I missed something that should affect my ratings.

**Typos Grammar Style And Presentation Improvements:**

I hope you can describe the dataset you use, if there is not enough space you can put it in the appendix.

---

> ### Author Rebuttal · Authors · 2023-08-29
>
> **Question-1: The abstract appears somewhat fragmented and lacks motivation**
>
> Thank you for bringing this to our attention, there is indeed better motivation for our work. Imagine that you are only allowed to change the last line of code while programming—how often would you need to start programming a function from scratch before it is correct? Auto-regressive models have a similar limitation while generating code: they do not allow the search to reconsider early tokens without a very expensive (grouped) beam search. By leveraging a diffusion process to gradually denoise a program based on a natural language condition, all program tokens can be reconsidered in each iteration. In combination with being conditioned on Gaussian noise, this allows us to generate more diverse candidate programs, while ensuring that all candidates are related to the condition.
>
> We will update our abstract to more closely reflect this intuition and motivation for applying diffusion to code.
>  \
>  \
>  \
> **Question-2: How does the design of the loss function, as described in line 171, ensure diversity of generated code?**
>
> The loss function itself is not what drives diversity, but it allows us to train the three main components of our model (denoiser, decoder and classification head) with a diffusion objective. It’s this diffusion objective, where the denoiser is conditioned on Gaussian noise, that drives diversity.
>
> Two of the three components of this loss were used in [1]. We add a term for the decoder, which we use to improve the syntactical correctness of generated code (see Question-6).
>
> We will clarify this in the revised version of the paper, as well as show results on the diversity of generated code.
> - Count of distinct token-level n-grams in the generated outputs divided by number of tokens [3].
> - Summary statistics over pairwise similarities of CodeBERT-based encoding of outputs [4].
> - Summary statistics over pairwise string edit distance of outputs [4].
>
> | Models     | 1-Gram | 2-Grams | 3-Grams | 4-Grams | Min Embedding Similarity | Max Embedding Similarity | Mean Embedding Similarity | Min Edit Distance | Max Edit Distance | Mean Edit Distance |
> |------------|--------|---------|---------|---------|--------------------------|--------------------------|---------------------------|-------------------|-------------------|--------------------|
> | T5         | 0.29   | 0.45    | 0.54    | 0.6     | 0.93                     | 0.99                     | 0.97                      | 0.08              | 0.58              | 0.36               |
> | CodeT5     | 0.27   | 0.41    | 0.46    | 0.51    | 0.94                     | 0.99                     | 0.97                      | 0.08              | 0.56              | 0.32               |
> | CodeFusion | 0.53   | 0.65    | 0.74    | 0.81    | 0.83                     | 0.97                     | 0.91                      | 0.21              | 0.83              | 0.57               |
>
> CodeFUSION’s higher n-gram fractions, lower embedding similarity, and higher edit distance on average support the claim that CodeFUSION generates more diverse programs.
>  \
>  \
>  \
> **Question-3: I recommend that you explore more large-scale code models such as CodeGen and CodeX.**
>
> We compared to text-davinci-003 (listed as GPT-3 in Table 1) and will clarify the exact endpoint in the paper. Text-davinci-003 supersedes CodeX, which has now been deprecated [2]. We have extended our evaluation to include CodeGen, along with three other recent models suggested by other reviewers (StarCoder, CodeT5+, ChatGPT/GPT-3.5). The updated evaluation table is shown below for Python (CoNaLa) Dataset.
>
> | System       | Model   | Param | Top-1 | Top-3 | Top-5 |
> |--------------|---------|-------|-------|-------|-------|
> | T5           | T5      | 770M  | 80.4  | 82.3  | 84.8  |
> | CodeT5       | T5      | 770M  | 80.5  | 83.1  | 85    |
> | GPT-3        | GPT 3   | 175B  | 82.5  | 83.7  | 85.8  |
> | ChatGPT      | GPT 3.5 | 20B   | 80.6  | 82.5  | 83.9  |
> | StarCoder    | GPT-2   | 15.5B | 79.2  | 82    | 84.1  |
> | CodeT5+      | T5      | 3B    | 78.6  | 81.1  | 83.5  |
> | CodeGen      | Custom  | 350M  | 80.1  | 81.8  | 83.7  |
> | Diffusion-LM | Custom  | 50M   | 70.4  | 74.3  | 76.5  |
> | GENIE        | Custom  | 93M   | 73.2  | 77.1  | 80.3  |
> | CODEFUSION   | T5      | 75M   | 80.7  | 86.3  | 90.3  |
>
>  \
> **Question-4: In Figure 1, how is the Gaussian noise added to the encoder? Why is there no connection between the Gaussian noise and the representations?**
>
> The noise is used by the denoiser, which takes in the embedded utterance (produced by the encoder) to condition the denoising. We will clarify this in the figure.
>  \
>  \
>  \
> **Question-5: I did not quite understand line 153. How does the formula in line 149 allow each token to be generated with full information about the other tokens?**
>
> The decoder is a transformer where each input token is allowed to attend to all other input tokens. Before mapping each embedded token back to a discrete token (with the classification head) we use this decoder to capture some syntactic properties of programs. For example, the denoiser can emit an embedded token that is similar to the representation for both “(“ and “)” tokens. The classification head can only look at one token and thus cannot reliably decide when the embedded tokens are very similar. The decoder, however, can look at all other tokens (through the full attention) and give more information to the classification head about which token to pick.
>  \
>  \
>  \
> **Question-6: Could you provide a more detailed explanation as to why this design can generate code with correct syntax and diversity?**
>
> This is partially answered in the previous question: a decoder with full attention prepares the denoised program tokens for being discretized. This decoder can be pre-trained without supervision on code snippets only, which teaches the denoiser, decoder and classification modules to generate syntactically correct programs. Diffusion helps in generating more diverse candidates by (1) being conditioned on Gaussian noise, and (2) allowing each token to be reconsidered during the (reverse) diffusion process.
>
> We will extend our discussion in the paper on this topic. Specifically, we find that pre-training the decoder on code-only helps generate syntactically valid code (see table below),  while the diffusion process induces diverse generations (see answer 2).
>
> |     System          |     Python    |     Bash    |     CF Rule    |
> |---------------------|---------------|-------------|----------------|
> |     Diffusion-LM    |     19.5      |     40.4    |     74.3       |
> |     Genie           |     24.2      |     54.2    |     78.6       |
> |     CodeFusion      |     67.6      |     73.4    |     94.5       |
>
>  \
> **Question-6: Missing lots of references**
>
> We will extend our related work discussion to reflect new work . Specifically, we will add [3,4,5,6,7,8,9,10,11] to related works and add [15,16,17,18] as new baselines to the evaluation.
>
> Please see the link below for our source code. We include this for review purposes only. We will extend the appendix with more details on training and dataset. In addition, we will be publicly releasing our benchmark tasks and our model’s generations.
>
> [Source Code](https://drive.google.com/drive/folders/1iDFbsH4tYmEJI_UXDh7jSRAD0RGO58y2?usp=sharing)
>  \
>  \
>  \
>  \
> **References:**
>
> [1]: Dockhorn, Tim, Arash Vahdat, and Karsten Kreis. "Genie: Higher-order denoising diffusion solvers." Advances in Neural Information Processing Systems 35 (2022): 30150-30166.
>
> [2]: https://platform.openai.com/docs/guides/code
>
> [3]: Vijayakumar, Ashwin K., et al. "Diverse beam search: Decoding diverse solutions from neural sequence models." arXiv preprint arXiv:1610.02424 (2016).
>
> [4]: Perlitz, Yotam, et al. "Diversity enhanced table-to-text generation via type control." arXiv preprint arXiv:2205.10938 (2022).
>
> [5]: Lomshakov, V., Kovalchuk, S., Omelchenko, M., Nikolenko, S., Aliev, A. (2023). Fine-Tuning Large Language Models for Answering Programming Questions with Code Snippets. In: Mikyška, J., de Mulatier, C., Paszynski, M., Krzhizhanovskaya, V.V., Dongarra, J.J., Sloot, P.M. (eds) Computational Science – ICCS 2023. ICCS 2023. Lecture Notes in Computer Science, vol 14074. Springer, Cham. https://doi.org/10.1007/978-3-031-36021-3_15
>
> [6]: Soliman, Ahmed S., Mayada M. Hadhoud, and Samir I. Shaheen. "MarianCG: A code generation transformer model inspired by machine translation." Journal of Engineering and Applied Science 69.1 (2022): 1-23.
>
> [7]: Beau, Nathanaël, and Benoît Crabbé. "The impact of lexical and grammatical processing on generating code from natural language." arXiv preprint arXiv:2202.13972 (2022).
>
> [8]: Norouzi, Sajad, Keyi Tang, and Yanshuai Cao. "Code Generation from Natural Language with Less Prior and More Monolingual Data." arXiv preprint arXiv:2101.00259 (2021).
>
> [9]: A Systematic Evaluation of Large Language Models of Code https://arxiv.org/abs/2202.13169
>
> [10]: A Survey on Pretrained Language Models for Neural Code Intelligence https://arxiv.org/abs/2212.10079
>
> [11]:	An Empirical Comparison of Pre-Trained Models of Source Code https://arxiv.org/abs/2302.04026
>
> [12]: Coder Reviewer Reranking for Code Generation https://arxiv.org/abs/2211.16490
>
> [13]:	TransCoder: Towards Unified Transferable Code Representation Learning Inspired by Human Skills https://arxiv.org/abs/2306.07285
>
> [14]:	WizardCoder: Empowering Code Large Language Models with Evol-Instruct https://arxiv.org/abs/2306.08568
>
> [15]:	Nijkamp, Erik, et al. "Codegen: An open large language model for code with multi-turn program synthesis." arXiv preprint arXiv:2203.13474 (2022).
>
> [16]: Li, Raymond, et al. "StarCoder: may the source be with you!." arXiv preprint arXiv:2305.06161 (2023).
>
> [17]:	Wang, Yue, et al. "Codet5+: Open code large language models for code understanding and generation." arXiv preprint arXiv:2305.07922 (2023).
>
> [18]:	https://openai.com/blog/chatgpt

---

### Official Review · Reviewer_9CNw · 2023-08-04

**Typos Grammar Style And Presentation Improvements:** There is a much room to improve the p…
**Soundness:** 3

**Excitement:**

3: Ambivalent: It has merits (e.g., it reports state-of-the-art results, the idea is nice), but there are key weaknesses (e.g., it describes incremental work), and it can significantly benefit from another round of revision. However, I won't object to accepting it if my co-reviewers champion it.

**Missing References:**

Some recent code LLMs such as CodeT5+ and StarCoder.

**Paper Topic And Main Contributions:**

This paper presents CodeFusion, the first diffusion model for code generation tasks. While the idea of exploring diffusion models for code is interesting and novel, this paper does not provide enough evidence for why diffusion models are better than conventional T5 or GPT-style code LLMs. For the design of CodeFusion, the authors insert a denoiser module to connect an encoder and decoder and claim that such a denoiser (acts as a diffusion module) is able to boost the generation diversity and leads to better code generation performance. Besides, this paper introduces another pretraining task of continuous paragraph denoising, which is parallel to the diffusion model design and claimed to improve the syntactic correctness of the generated programs. The experiments show that CodeFusion achieves better results on 3 code generation benchmarks.

However, this denoiser module also introduces extra parameters compared to a vanilla encoder-decoder model, making it difficult to determine whether the performance improvement comes from the novel design or just from the large model size. Furthermore, there are no experimental results for analyzing the better diversity of the generated programs (showing better top-3 and top-5 results in Table 1 does not imply better diversity). Worse still, this paper does not compare with SoTA code LLMs such as StarCoder or CodeT5+ and more popular code generation benchmarks such as HumanEval and APPS.

**Questions For The Authors:**

* In Table 2, why don't you compare with a stronger baseline such as CodeT5 which is included for comparison in Table 1?
* It would be better to separate the effects of the diffusion module and CPD pretraing tasks, which are parallel to each other. Sometimes their effects are mixed and coupled in the paper presentation. Can you explain the effects for each of them?

**Reasons To Accept:**

* Novel and interesting idea to explore diffusion models for code generation

**Reasons To Reject:**

* The motivation of the diffusion model for better diversity is not convincing.  There are no experimental results for analyzing the better diversity of the generated programs In Table 1, showing better top-3 and top-5 results does not imply better diversity. For better diversity, a commonly adopted approach is to use samping decoding with a higher temperature. This work needs to be compared with this to better showcase its benefit of high diverse generations.
* The experimental results cannot validate the advantages of CodeFusion over other code LLMs. In Table 1, especially for the top-1 results, the performance gains between CodeFusion and CodeT5 are quite limited, even given that CodeFusion has a large model size by introducing a denoiser module consisting of 10 Transformer layers (the Param column should be updated with real total parameter counts for CodeFusion).
* This work lacks comparison to other SoTA code LLMs such as StarCoder or CodeT5+ and more popular code generation benchmarks such as HumanEval and APPS.

**Reproducibility:**

3: Could reproduce the results with some difficulty. The settings of parameters are underspecified or subjectively determined; the training/evaluation data are not widely available.

**Reviewer Confidence:**

4: Quite sure. I tried to check the important points carefully. It's unlikely, though conceivable, that I missed something that should affect my ratings.

---

> ### Author Rebuttal · Authors · 2023-08-29
>
> **Question-1: There are no experimental results for analyzing the better diversity of the generated programs in Table 1**
>
> Thank you for making this observation. We will extend the next version of the paper with the table below. This table details the following for CodeFUSION and other models:
> - Count of distinct token-level n-grams in the generated outputs divided by number of tokens [1].
> - Summary statistics over pairwise similarities of CodeBERT-based encoding of outputs [2].
> - Summary statistics over pairwise string edit distance of outputs [2].
>
> |     Models        |     1-Gram    |     2-Grams    |     3-Grams    |     4-Grams    |     Min Embedding Similarity    |     Max Embedding Similarity    |     Mean Embedding Similarity    |     Min Edit Distance    |     Max Edit Distance    |     Mean Edit Distance    |
> |-------------------|---------------|----------------|----------------|----------------|---------------------------------|---------------------------------|----------------------------------|--------------------------|--------------------------|---------------------------|
> |     T5            |     0.29      |     0.45       |     0.54       |     0.6        |     0.93                        |     0.99                        |     0.97                         |     0.08                 |     0.58                 |     0.36                  |
> |     CodeT5        |     0.27      |     0.41       |     0.46       |     0.51       |     0.94                        |     0.99                        |     0.97                         |     0.08                 |     0.56                 |     0.32                  |
> |     CodeFusion    |     0.53      |     0.65       |     0.74       |     0.81       |     0.83                        |     0.97                        |     0.91                         |     0.21                 |     0.83                 |     0.57                  |
>
>  \
> CodeFUSION’s higher n-gram fractions, lower embedding similarity, and higher edit distance on average support the claim that CodeFUSION generates more diverse programs.
>  \
>  \
>  \
> **Question-2: CodeFUSION has a large model size by introducing a denoiser module**
>
> Thank you for catching this mistake– Table 1 model size for CodeFUSION is not correctly detailed (this was the result of a copy error).  Please see the corrected model sizes below.
>
> |     Model           |     Size     |
> |---------------------|--------------|
> |     T5              |     770M     |
> |     CodeT5          |     770M     |
> |     GPT-3           |     175B     |
> |     StarCoder       |     15.5B    |
> |     CodeGen         |     350M     |
> |     CodeT5+         |     16B      |
> |     Diffusion-LM    |     50M      |
> |     Genie           |     93M      |
> |     CodeFusion      |     75M      |
>
> CodeT5 is roughly 10 times the size of CodeFUSION.
>  \
>  \
>  \
> **Question-3: This work lacks comparison to other SoTA code LLMs like StarCoder and CodeT5+ and more popular code generation benchmarks such as HumanEval and APPS.**
>
> We have extended our evaluation to include CodeT5+ [9], StarCoder [8], ChatGPT [10], and CodeGen [7] as suggested by other reviewers—thank you for your suggestion. We will also detail some interesting future work based on this. The table below summarizes their performance. Note that all other models have significantly more parameters. CodeT5+ is 40 times larger, while StarCoder is 200 times larger than CodeFUSION.
>
> We used two previously published benchmarks for NL to Code in Python and Bash. The Python benchmark, CoNaLa, has been used in [3,4,5,6]. We chose these benchmarks as the average code length (number of tokens) is 15.7 compared to 74.3 in HumanEval/APPs. Extending CodeFUSION to longer code fragments remains future work and we will discuss this limitation in the next version of the paper.
>
> The updated evaluation table is shown below for Python (CoNaLa) Dataset.
>
> |     System          |     Model      |     Param    |     Top-1    |     Top-3    |     Top-5    |
> |---------------------|----------------|--------------|--------------|--------------|--------------|
> |     T5              |     T5         |     770M     |     80.4     |     82.3     |     84.8     |
> |     CodeT5          |     T5         |     770M     |     80.5     |     83.1     |     85       |
> |     GPT-3           |     GPT 3      |     175B     |     82.5     |     83.7     |     85.8     |
> |     ChatGPT         |     GPT 3.5    |     20B      |     80.6     |     82.5     |     83.9     |
> |     StarCoder       |     GPT-2      |     15.5B    |     79.2     |     82       |     84.1     |
> |     CodeT5+         |     T5         |     16B      |     79.6     |     82.1     |     84.5     |
> |     CodeGen         |     Custom     |     350M     |     80.1     |     81.8     |     83.7     |
> |     Diffusion-LM    |     Custom     |     50M      |     70.4     |     74.3     |     76.5     |
> |     GENIE           |     Custom     |     93M      |     73.2     |     77.1     |     80.3     |
> |     CODEFUSION      |     T5         |     75M      |     80.7     |     86.3     |     90.3     |
>
>  \
> **Questtion-4: In Table 2, why don’t you compare with a stronger baseline such as CodeT5 which is included for comparison in Table 1?**
>
> The scope of table 2 is to compare CodeFUSION with existing text diffusion models to highlight that adding the decoder helps CodeFUSION in generating syntactically correct code. We will clarify this in the next version.
>
> |     System          |     Python    |     Bash    |     CF Rule    |
> |---------------------|---------------|-------------|----------------|
> |     Diffusion-LM    |     19.5      |     40.4    |     74.3       |
> |     Genie           |     24.2      |     54.2    |     78.6       |
> |     CodeFusion      |     67.6      |     73.4    |     94.5       |
>
>  \
> **Question-5: It would be better to separate the effects of the diffusion module and CPD pretraining tasks.**
>
> Without the diffusion module, CodeFUSION essentially becomes CodeT5. An ablation on removing CPD can be found in Table 2 (shown as M – CPD objective), as well as CodeFUSION without the decoder (shown as M with Grounding and M with clamping). We will improve the clarity of this table.
>  \
>  \
>  \
> **Reproducibility and Benchmarks**
>
> Please see the link below for our source code. We include this for review purposes only. We will extend the appendix with more details on training and dataset. In addition, we will be publicly releasing our benchmark tasks and our model’s generations.
>
> [Source Code](https://drive.google.com/drive/folders/1iDFbsH4tYmEJI_UXDh7jSRAD0RGO58y2?usp=sharing)
>  \
>  \
>  \
> **References:**
>
> [1]: Vijayakumar, Ashwin K., et al. "Diverse beam search: Decoding diverse solutions from neural sequence models." arXiv preprint arXiv:1610.02424 (2016).
>
> [2]: Perlitz, Yotam, et al. "Diversity enhanced table-to-text generation via type control." arXiv preprint arXiv:2205.10938 (2022).
>
> [3]: Lomshakov, V., Kovalchuk, S., Omelchenko, M., Nikolenko, S., Aliev, A. (2023). “Fine-Tuning Large Language Models for Answering Programming Questions with Code Snippets.” ICCS 2023.
>
> [4]: Soliman, Ahmed S., Mayada M. Hadhoud, and Samir I. Shaheen. "MarianCG: A code generation transformer model inspired by machine translation." Journal of Engineering and Applied Science 69.1 (2022): 1-23.
>
> [5]: Beau, Nathanaël, and Benoît Crabbé. "The impact of lexical and grammatical processing on generating code from natural language." ACL 2022.
>
> [6]: Norouzi, Sajad, Keyi Tang, and Yanshuai Cao. "Code Generation from Natural Language with Less Prior and More Monolingual Data." arXiv preprint arXiv:2101.00259 (2021).
>
> [7]: Nijkamp, Erik, et al. "Codegen: An open large language model for code with multi-turn program synthesis." arXiv preprint arXiv:2203.13474 (2022).
>
> [8]: Li, Raymond, et al. "StarCoder: may the source be with you!." arXiv preprint arXiv:2305.06161 (2023).
>
> [9]: Wang, Yue, et al. "Codet5+: Open code large language models for code understanding and generation." arXiv preprint arXiv:2305.07922 (2023).
>
> [10]: https://openai.com/blog/chatgpt

---

### Meta-Review · Area_Chair_P7MC · 2023-09-18

**Recommendation:** 4

**Metareview:**

This paper introduces a diffusion model for natural language to code generation problems. Code generation is a growing area of research and this paper adds the first diffusion model to improve the diversity of the generated code. Reviewers find that the presentation is clear, the method to be novel and interesting. Reviewers express concern on lack of evidence for why diffusion models work better, for which authors responded with fine-grained analysis of diversity of generated programs. Authors also added the requested baselines such as StartCoder and CodeT5. Authors should add these results along with other clarifications in the revision.

---

### Decision · Program_Chairs · 2023-10-07

**Decision:**

Accept-Main

**Comment:**

This paper introduces a diffusion model for natural language to code generation problems. Code generation is a growing area of research and this paper adds the first diffusion model to improve the diversity of the generated code. Reviewers find that the presentation is clear, the method to be novel and interesting. Reviewers express concern on lack of evidence for why diffusion models work better, for which authors responded with fine-grained analysis of diversity of generated programs. Authors also added the requested baselines such as StartCoder and CodeT5. Authors should add these results along with other clarifications in the revision.